# Increased risk of hospitalisation and intensive care admission associated with reported cases of SARS-CoV-2 variants B.1.1.7 and B.1.351 in Norway, December 2020–May 2021

**Lamprini Veneti**[1], **Elina Seppälä**[2,3], **Margrethe Larsdatter Storm**[4], **Beatriz Valcarcel Salamanca**[5], **Eirik Alnes Buanes**[6,7], **Nina Aasand**[4], **Umaer Naseer**[1], **Karoline Bragstad**[8], **Olav Hungnes**[8], **Håkon Bøås**[1], **Reidar Kvåle**[6,9], **Karan Golestani**[1], **Siri Feruglio**[1], **Line Vold**[1], **Karin Nygård**[1], **Robert Whittaker**[2]*

1 Department of Infection Control and Preparedness, Norwegian Institute of Public Health, Oslo, Norway, 2 Department of Infection Control and Vaccines, Norwegian Institute of Public Health, Oslo, Norway, 3 European Programme for Intervention Epidemiology Training (EPIET), European Centre for Disease Prevention and Control (ECDC), Solna, Sweden, 4 Department of Infectious Disease Registries, Norwegian Institute of Public Health, Oslo, Norway, 5 Department of Method Development and Analytics, Norwegian Institute of Public Health, Oslo, Norway, 6 Department of Anaesthesia and Intensive Care, Haukeland University Hospital, Bergen, Norway, 7 Norwegian Intensive Care and Pandemic Registry, Haukeland University Hospital, Bergen, Norway, 8 Department of Virology, Norwegian Institute of Public Health, Oslo, Norway, 9 Department of Clinical Medicine, University of Bergen, Bergen, Norway

* Robert.Whittaker@fhi.no

## Abstract

### Introduction

Since their emergence, SARS-CoV-2 variants of concern (VOC) B.1.1.7 and B.1.351 have spread worldwide. We estimated the risk of hospitalisation and admission to an intensive care unit (ICU) for infections with B.1.1.7 and B.1.351 in Norway, compared to infections with non-VOC.

### Materials and methods

Using linked individual-level data from national registries, we conducted a cohort study on laboratory-confirmed cases of SARS-CoV-2 in Norway diagnosed between 28 December 2020 and 2 May 2021. Variants were identified based on whole genome sequencing, partial sequencing by Sanger sequencing or PCR screening for selected targets. The outcome was hospitalisation or ICU admission. We calculated adjusted risk ratios (aRR) with 95% confidence intervals (CIs) using multivariable binomial regression to examine the association between SARS-CoV-2 variants B.1.1.7 and B.1.351 with i) hospital admission and ii) ICU admission compared to non-VOC.

### Results

We included 23,169 cases of B.1.1.7, 548 B.1.351 and 4,584 non-VOC. Overall, 1,017 cases were hospitalised (3.6%) and 206 admitted to ICU (0.7%). B.1.1.7 was associated with a 1.9-fold increased risk of hospitalisation (aRR 95%CI 1.6–2.3) and a 1.8-fold

**Data Availability Statement:** The dataset analysed in the study contains individual-level linked data from various central health registries, national clinical registries and other national administrative registries in Norway. The researchers had access to the data through the national emergency preparedness registry for COVID-19 (Beredt C19), housed at the Norwegian Institute of Public Health (NIPH). In Beredt C19, only fully anonymised data (i.e. data that are neither directly nor potentially indirectly identifiable) are permitted to be shared publicly. Legal restrictions therefore prevent the researchers from publicly sharing the dataset used in the study that would enable others to replicate the study findings. However, external researchers are freely able to request access to linked data from the same registries from outside the structure of Beredt C19, as per normal procedure for conducting health research on registry data in Norway. Further information on Beredt C19, including contact information for the Beredt C19 project manager, and information on access to data from each individual data source, is available at https://www.fhi.no/en/id/infectious-diseases/coronavirus/emergency-preparedness-register-for-covid-19/.

**Funding:** The authors received no specific funding for this work.

**Competing interests:** The authors have declared that no competing interests exist.

increased risk of ICU admission (aRR 95%CI 1.2–2.8) compared to non-VOC. Among hospitalised cases, no difference was found in the risk of ICU admission between B.1.1.7 and non-VOC. B.1.351 was associated with a 2.4-fold increased risk of hospitalisation (aRR 95%CI 1.7–3.3) and a 2.7-fold increased risk of ICU admission (aRR 95%CI 1.2–6.5) compared to non-VOC.

## Discussion

Our findings add to the growing evidence of a higher risk of severe disease among persons infected with B.1.1.7 or B.1.351. This highlights the importance of prevention and control measures to reduce transmission of these VOC in society, particularly ongoing vaccination programmes, and preparedness plans for hospital surge capacity.

## Introduction

Multiple variants of SARS-CoV-2, the causative agent of COVID-19, have been observed worldwide. Based on evidence regarding increased transmissibility, disease severity and/or ability to evade immunity generated during previous infection or vaccination, variants of concern (VOC) have been identified. These include lineages B.1.1.7 (alpha variant), first detected in the United Kingdom (UK), and B.1.351 (beta variant), first detected in South Africa. Since their emergence, both variants have spread worldwide [1].

In Norway (population 5.4 million), testing activity for COVID-19 is high, with consistently over 100,000 persons tested weekly (defined as one or more tests per person within a seven-day period) since week 44, 2020, while mathematical modelling has estimated that consistently over 50% of all cases weekly have been diagnosed since late 2020 [2]. Sequencing capacity in Norwegian laboratories was rapidly scaled up from early December 2020, and the capacity to screen for variants or perform whole genome sequencing (WGS) was further increased following reports of widespread transmission of B.1.1.7 in the UK. According to the Norwegian national laboratory database, the proportion of all laboratory-confirmed cases of COVID-19 in Norway who had available data on the variant of SARS-CoV-2 that caused their infection increased from 6% in week 53/2020 to 80% in week 9/2021. The first infection with B.1.1.7 was sampled in week 48/2020. B.1.1.7 was the variant detected in at least 50% of cases from week 5 (including over 90% weekly from week 13 to week 20), until week 27/2021, when the VOC B.1.617.2 (delta variant) began to predominate. The first infection with B.1.351 was sampled in week 53/2020 and the variant has caused several outbreaks in the country [2].

In Norway, increasing detection of B.1.1.7 and B.1.351 in early 2021 coincided with a rapid increase in the number of new admissions to hospital and intensive care units (ICU) among confirmed cases of COVID-19. Simultaneously, unadjusted surveillance data signalled increased risk of hospitalisation, especially among younger and middle-aged adults [3]. During this period, the hospital system functioned within capacity and hospital treatment was available to all patients who would benefit. Hospital admission criteria were consistent. A few studies have indicated that B.1.1.7 infection is associated with increased risk of hospitalisation, admission to ICU and/or death [4–11]. For B.1.351, a study using pooled surveillance data from seven European countries found infection with this variant to be associated with higher odds of hospitalisation, as well as ICU admission among those aged 40–59 years [7]. Also, data from South Africa have suggested increased risk of acute severe disease on admission [12] and increased mortality among hospitalised patients [13] during the second wave, when B.1.351 dominated.

Using linked individual-level data, we estimated the risk of hospitalisation and ICU admission for infections with B.1.1.7 and B.1.351 in Norway compared to infections with other variants (non-VOC), after accounting for demographic characteristics and underlying comorbidities.

## Materials and methods

### Study design and setting

We conducted a cohort study, including cases who tested positive for SARS-CoV-2 in Norway between 28 December 2020 and 2 May 2021, who had a national identity number registered and who had available virus variant data after screening with PCR or WGS. Cases that were vaccinated against COVID-19 with at least one dose before diagnosis or had been hospitalised with COVID-19 not reported as the main cause of admission were excluded. We extracted data from the national emergency preparedness register, Beredt C19, up to 8 June 2021, ensuring a minimum of 36 days follow-up since last sampling date.

### Data sources

The national emergency preparedness register contains individual-level data from central health registries, national clinical registries and other national administrative registries [14]. We included data on notified cases of laboratory-confirmed SARS-CoV-2 infection from the Norwegian Surveillance System for Communicable Diseases (MSIS). We obtained data on hospitalisation and intensive care admission following a positive SARS-CoV-2 test from the Norwegian Intensive Care and Pandemic Registry (NIPaR). All Norwegian hospitals report to NIPaR and reporting is mandatory. Data on virus variants came from the MSIS laboratory database (national laboratory database), which receives SARS-CoV-2 test results from all Norwegian microbiology laboratories. Data on underlying comorbidities, as stipulated by the national COVID-19 vaccination programme [15], was based on ICD-10 codes from the Norwegian Patient Registry and ICPC-2 codes from the Norway Control and Payment of Health Reimbursement database (S1 File, part 1). Data on COVID-19 vaccinations came from the Norwegian Immunisation Registry, SYSVAK. Data on persons with a national identity number was drawn from the national population registry. The national identity number was essential to link data from all registries used in the analysis. More detailed information on each data source can be found at [14].

### Laboratory investigations and classification of variants

Variants were identified based on WGS using Illumina or Nanopore technology, partial sequencing by Sanger sequencing or PCR screening for selected targets. PCR screening methods include real-time RT-PCR, sometimes in combination with melting curve analysis, of one or several mutation targets specific to the different variants. Variants B.1.1.7 and B.1.351 were assigned based on the identified specific PCR targets or, in addition, if available, to pangolin lineage after sequencing. Other VOC (P.1 and B.1.617.2), as defined by the European Centre for Disease Prevention and Control on 3 June 2021 [16], and cases for whom VOC and non-VOC could not clearly be distinguished were excluded. All other variants were classified as non-VOC. Cases of B.1.1.7, B.1.351 and non-VOC are henceforth referred to collectively as our 'study cohort'. The distribution of different variants by week is presented in S1 File, part 2.

### Hospitalisation and ICU admission

Hospitalisation was defined as hospitalisation following a positive SARS-CoV-2 test, where COVID-19 was reported as the main cause of admission. Cases hospitalised with other or

unknown main cause of admission were excluded from the study population in order to avoid bias. ICU admission was defined as admission to ICU following a positive SARS-CoV-2 test. All admissions to hospital and ICU were included, regardless of the length of stay.

## Data analysis

We described cases in terms of demographic characteristics, number of underlying comorbidities, microbiological characteristics, hospital and ICU admission. We assessed the representativeness of our study population by comparing the characteristics of our study cohort and notified cases using chi-square tests (S1 File, part 3.1).

**Hospitalisation.** We used Wilcoxon rank-sum tests to test differences in the distribution of i) time between symptom onset and date of first hospitalisation ii) sampling date and date of first hospitalisation and iii) length of stay in hospital between B.1.1.7, B.1.351 and non-VOC.

We calculated adjusted risk ratios (aRR) with 95% confidence intervals (CIs) using multivariable binomial regression to examine the association between SARS-CoV-2 variants B.1.1.7 and B.1.351 with hospital admission compared to non-VOC (main analysis). Co-variates considered in the model selection were age (4 age groups), gender, country of birth (3 levels), period of sampling (biweekly), county of residence (12 levels) and number underlying comorbidities (0, 1, ≥2). Model selection for each multivariable regression was conducted using the likelihood ratio test and the Akaike Information Criterion. We checked for interactions between our co-variates by including interaction terms in our models. Similarly, we estimated the adjusted odds ratios (aOR) of hospitalisation for B.1.1.7 cases compared to B.1.351 using logistic regression, as binomial regression did not converge in this analysis.

In addition to our main analysis, we conducted a number of sensitivity analyses by extending or restricting our study population (for example including only cases who had WGS results) and by adjusting our outcome definitions (for example, including all cases who were hospitalised regardless of main cause of admission) to further explore if our main results were robust (S1 File, part 3.2).

In S1 File, part 3.3 we also present the results from a univariate and multivariable logistic regression adjusted for variant (categorical variable with three levels), sex, age group, country of birth and number of underlying comorbidities. This allows readers to see the estimates for hospitalisation for other co-variates that were included in the multivariable analysis. Logistic regression was used in this analysis since binomial regression did not converge.

**ICU admission.** As in our main analysis, we calculated the aRR of ICU admission for cases with B.1.1.7 or B.1.351 compared to non-VOC. We also calculated the aRR of ICU admission for B.1.1.7 cases compared to B.1.351. After restricting our study population to hospitalised cases, we also calculated the aRR of ICU admission for hospitalised cases with B.1.1.7 or B.1.351 compared to non-VOC. In these analyses we adjusted for the same variables as in our main analysis for hospitalisation, regardless of whether all included variables were significant (county of residence and period of sampling were also checked but were not added since they were not significant).

Statistical significance was considered to be a p value <0.05. Statistical analysis was performed in Stata version 16 (Stata Corporation, College Station, Texas, US).

## Ethics

Ethical approval for this study was granted by Regional Committees for Medical Research Ethics—South East Norway, reference number 249509. The need for informed consent was waived by the ethics committee.

## Results

### Description of cohort

During the study period, 65,040 laboratory-confirmed cases of COVID-19 were reported. Of these, we excluded 2,138 who did not have a national identity number and a further 1,902 who were vaccinated before diagnosis. We excluded an additional 383 cases hospitalised with other main cause of admission than COVID-19 and 11 with unknown main cause of admission. From the remaining 60,606 notified cases (93%), 29,979 (49%) had known variant data. The proportion of cases with known variant data increased from 8% in weeks 53/2020–1/2021 to 78% in weeks 8–9 and stayed between 46 to 75% during weeks 10–17 (Fig 1).

Of these 29,979, there were seven cases of P.2 and 12 B.1.617.2, while for 1,659 cases it was not possible to distinguish between B.1.1.7, B.1.351, other VOC and non-VOC. The remaining 28,301 cases of B.1.1.7, B.1.351 and non-VOC were included in our study cohort.

We found differences between our study cohort and notified cases with regards to county of residence, sampling week, age, number of comorbidities, and hospitalisation (S1 File, part 3.1). Differences in county and sampling week reflect the evolution of the outbreak as well as the introduction of PCR screening methodology for virus variants at primary diagnostic laboratories. A lower proportion of cases in the age group above 65 years were included in our study cohort compared to other age groups (41% vs 46–47%). A lower proportion of cases with two or more comorbidities were included compared to cases with one or no comorbidities (43% vs 46–47%). The differences in county of residence, sampling week, age, number of comorbidities were considered minor for our study design and aim. Among hospitalised cases 54% were included in our study cohort, while among non-hospitalised cases 46% were included. We conducted a sensitivity analysis which indicated that the 8% difference between those hospitalised and not hospitalised did not influence our estimates in our main analysis as presented below (S1 File, part 3.2).

Among the 28,301 cases in our study cohort, WGS was performed on 12,747 (45%) and 15,554 (55%) were screened by PCR. Based on the results from both methods, 23,169 (77%) cases were classified as variant B.1.1.7 (39% of those through WGS), 548 (2%) variant B.1.351 (81% WGS) and 4,584 (15%) as non-VOC (69% WGS). The proportion of cases with B.1.1.7 infection increased from 13% in weeks 53/2020–1/2021 to 94% in weeks 10–11/2021 and reached 99% in weeks 16–17. During the same period, cases with B.1.351 infection increased from 0.3% in weeks 53–1 to 3.8% in weeks 6–7 and gradually decreased to 0.2% in weeks 16–17 (Fig 1). There was a lower proportion of cases of B.1.1.7 and B.1.351 among those aged 65 years and over, compared to the other age groups and a lower proportion of cases of B.1.1.7 among those with at least two comorbidities compared to those with less than two comorbidities (Table 1).

### Hospitalisation

Overall, 1,017 (3.6%) cases in our study cohort were hospitalised with COVID-19 as the main cause of hospitalisation. The proportion of cases hospitalised was 3.8% (884) for B.1.1.7, 4.2% (23) for B.1.351, and 2.4% (110) for non-VOC. Data on date of symptom onset were available for 500 patients (49%). The median time from symptom onset to hospitalisation was 8 days (IQR: 6–11). For all 1,017 patients, the median time from testing to hospitalisation was 6 days (IQR: 3–9) and the median length of stay in hospital was 5 days (IQR: 3–10). We observed no difference between the three variant groups for these three parameters (Wilcoxon rank-sum tests not significant). For all variants, the proportion of cases hospitalised increased with age, number of comorbidities and was higher among men and foreign-born (Table 1).

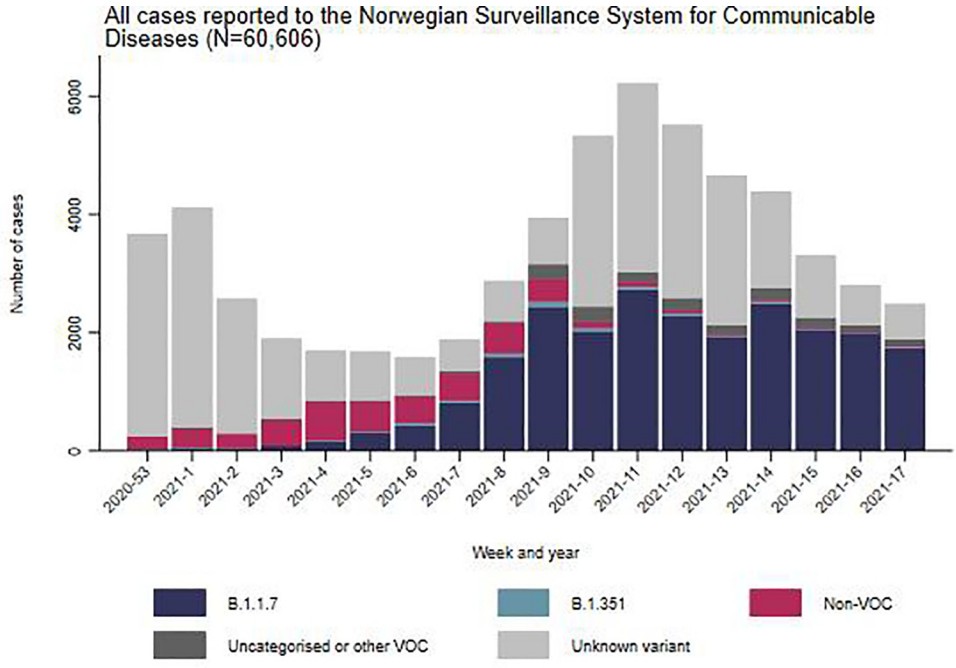

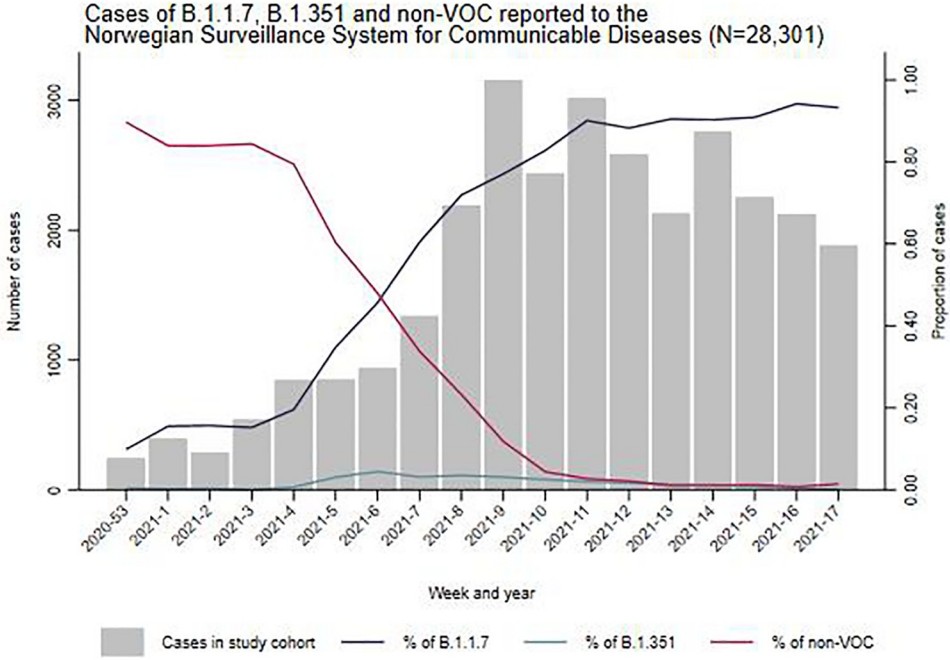

**Fig 1. Distribution of COVID-19 cases reported and screened with B.1.1.7, B.1.351 and non-VOC by week of sampling, Norway, December 2020—May 2021. Note:** 'All cases reported' (n = 60,606) excludes vaccinated cases, cases who did not have a national identity number registered and hospitalised cases who did not have COVID-19 as their main cause of admission (n = 4,434). 'Uncategorised or other variant of concern' (VOC) includes cases of P.2 (n = 7) and B.1.617.2 (n = 12), and cases for whom B.1.1.7, B.1.351, other VOC and non-VOC could not clearly be distinguished (n = 1,659).

**Table 1. Distribution of cases in study cohort by detected variants for different characteristics and proportion hospitalised, Norway, December 2020—May 2021.**

| Characteristics | | Study cohort | Variant type (% of study cohort) | | | Hospitalised cases (%) | | | |
|---|---|---|---|---|---|---|---|---|---|
| | | | Non-VOC | B.1.1.7 | B.1.351 | Total | Non-VOC | B.1.1.7 | B.1.351 |
| **Total** | | 28,301 (100%) | 4,584 (16%) | 23,169 (82%) | 548 (1.9%) | 1,017 (3.6%) | 110 (2.4%) | 884 (3.8%) | 23 (4.2%) |
| **Sex** | Female | 13,292 (47%) | 2,134 (16%) | 10,902 (82%) | 256 (1.9%) | 405 (3.1%) | 39 (1.8%) | 357 (3.3%) | 9 (3.5%) |
| | Male | 15,009 (53%) | 2,450 (16%) | 12,267 (82%) | 292 (2.0%) | 612 (4.1%) | 71 (2.9%) | 527 (4.3%) | 14 (4.8%) |
| **Age group** | 0–24 years | 11,782 (42%) | 1,636 (14%) | 9,915 (84%) | 231 (2.0%) | 41 (0.4%) | 5 (0.3%) | 36 (0.4%) | 0 (0%) |
| | 25–44 years | 9,370 (33%) | 1,561 (17%) | 7,624 (81%) | 185 (2.0%) | 236 (2.5%) | 14 (0.9%) | 217 (2.9%) | 5 (2.7%) |
| | 45–64 years | 6,103 (22%) | 1,125 (18%) | 4,860 (80%) | 118 (1.9%) | 496 (8.1%) | 50 (4.4%) | 431 (8.9%) | 15 (13%) |
| | ≥65 years | 1,046 (3.7%) | 262 (25%) | 770 (74%) | 14 (1.3%) | 244 (23%) | 41 (16%) | 200 (26%) | 3 (21%) |
| **Norwegian born** | Yes | 17,865 (63%) | 2,991 (17%) | 14,547 (81%) | 327 (1.8%) | 478 (2.7%) | 64 (2.1%) | 405 (2.8%) | 9 (2.8%) |
| | No | 10,186 (36%) | 1,554 (15%) | 8,415 (83%) | 217 (2.1%) | 514 (5.1%) | 41 (2.6%) | 459 (5.5%) | 14 (6.5%) |
| | Unknown | 250 (0.9%) | 39 (16%) | 207 (83%) | 4 (1.6%) | 25 (10%) | 5 (13%) | 20 (9.7%) | 0 (0%) |
| **Number of comorbidities** | 0 | 24,863 (88%) | 3,953 (16%) | 20,430 (82%) | 480 (1.9%) | 597 (2.4%) | 52 (1.3%) | 532 (2.6%) | 13 (2.7%) |
| | 1 | 2,940 (10%) | 510 (17%) | 2,372 (81%) | 58 (2.0%) | 293 (10%) | 34 (6.7%) | 253 (11%) | 6 (10%) |
| | ≥2 | 498 (1.8%) | 121 (24%) | 367 (74%) | 10 (2.0%) | 127 (26%) | 24 (20%) | 99 (27%) | 4 (40%) |

In the univariate analyses, B.1.1.7 was associated with 1.6-fold increased risk of hospitalisation (RR 95%CI 1.3–1.9), and B.1.351 with a 1.7-fold increased risk of hospitalisation (RR 95% CI 1.1–2.7) compared to non-VOC. In Table 2, we present the estimated association between

**Table 2. Association between hospitalisation and infection with B.1.1.7 and B.1.351 compared to non-VOC of SARS-CoV-2 (adjusted for demographic characteristics and number of comorbidities) overall and stratified by sex, age groups, country of origin, and number of comorbidities, Norway, December 2020—May 2021.**

| | | Infection with B.1.1.7 | | | Infection with B.1.351 | | |
|---|---|---|---|---|---|---|---|
| | | Hospitalisation | | | Hospitalisation | | |
| | | No | Yes (%) | Adjusted RR * (95% CI) | No | Yes (%) | Adjusted RR ** (95% CI) |
| **Overall** | Non-VOC | 4,474 | 110 (2.4) | Ref | 4,474 | 110 (2.4) | Ref |
| | B.1.1.7 | 22,285 | 884 (3.8) | 1.95 (1.61–2.34) | - | - | - |
| | B.1.351 | - | - | - | 525 | 23 (4.2) | 2.37 (1.71–3.27) |
| **Stratified:** | | | | | | | |
| **By sex** | Female | 10,545 | 357 (3.3) | 2.03 (1.47–2.79) | 247 | 9 (3.5) | 2.71 (1.39–5.27) |
| | Male | 11,740 | 527 (4.3) | 1.90 (1.50–2.39) | 278 | 14 (4.8) | 2.46 (1.64–3.68) |
| **By age group** | 0–24 years | 9,879 | 36 (0.4) | 1.15 (0.46–2.90) | 231 | 0 (0.0) | 0.90 (0.10–7.98) *** |
| | 25–44 years | 7,407 | 217 (2.9) | 3.05 (1.78–5.21) | 180 | 5 (2.7) | 2.60 (0.93–7.24) |
| | 45–64 years | 4,429 | 431 (8.9) | 2.02 (1.52–2.67) | 103 | 15 (13) | 3.29 (2.18–4.95) |
| | >65 years | 570 | 200 (26) | 1.63 (1.20–2.21) | 11 | 3 (21) | 1.32 (0.50–3.51) |
| **By country of birth (Norwegian born)** | Yes | 14,142 | 405 (2.8) | 1.89 (1.47–2.42) | 318 | 9 (2.8) | 1.88 (0.97–3.65) |
| | No | 7,956 | 459 (5.5) | 2.20 (1.62–2.99) | 203 | 14 (6.5) | 3.03 (1.95–4.70) |
| | Unknown | 187 | 20 (9.7) | 0.92 (0.38–2.20) | 4 | 0 (0.0) | - |
| **By number of comorbidities** | 0 | 19,898 | 523 (2.6) | 2.32 (1.76–3.06) | 467 | 13 (2.7) | 2.47 (1.38–4.41) |
| | 1 | 2,119 | 253 (11) | 1.84 (1.32–2.58) | 52 | 6 (10) | 2.08 (0.93–4.69) |
| | > = 2 | 268 | 99 (27) | 1.48 (1.00–2.19) | 6 | 4 (40) | 2.50 (1.43–4.36) |

RR: risk ratio; CI: confidence intervals.

* Adjusted for sex, age group, country of birth and number of comorbidities. In this analysis the variant was included as a 2-level categorical variable B.1.1.7 and non-VOC. Week of sampling and county of residence were not significant predictors in the multivariable model.

** Adjusted for sex, age group, country of birth and number of comorbidities. In this analysis the variant was included as a 2-level categorical variable B.1.351 and non-VOC. Week of sampling and county of residence were not significant predictors in the multivariable model.

*** We assigned randomly one of the cases with B.1.351 as hospitalised in order to be able to calculate the aRR estimate in this stratum.

infection with B.1.1.7 and B.1.351 separately, compared to non-VOC infections and the aRRs of hospitalisation. In the multivariable analysis, after adjusting for sex, age group, country of birth and number of comorbidities, B.1.1.7 was associated with a 1.9-fold increased risk of hospitalisation (aRR 95%CI 1.6–2.3), and B.1.351 was associated with a 2.4-fold increased risk of hospitalisation (aRR 95%CI 1.7–3.3) compared to non-VOC. In Table 2, we also present the estimates for B.1.1.7 and B.1.351 variants compared to non-VOC stratified by sex, age group, country of birth and number of comorbidities. The aRR estimates for B.1.1.7 and B.1.351 for several of the strata indicate a similar association. In addition, our main results were robust in the several sensitivity analyses that we conducted (S1 File, part 3.2). No significant difference was found in the odds of hospitalisation for B.1.1.7 cases when compared to B.1.351 (aOR 1.2; 95%CI 0.7–1.8).

## ICU admission

Overall, 206 (0.7%) cases were admitted to ICU. The proportion admitted to ICU was 0.8% (176) for B.1.1.7, 0.9% (5) for B.1.351 and 0.6% (25) for non-VOC. Infection with B.1.1.7 was associated with a 1.8-fold increased risk of admission to ICU (aRR 95%CI 1.2–2.8) compared to non-VOC after adjusting for sex, age group, country of birth and comorbidities. Similarly, infection with B.1.351 was associated with a 2.7-fold increased risk of ICU admission (aRR 95%CI 1.2–6.5). No significant difference was found in the risk of ICU admission for B.1.1.7 cases when compared to B.1.351 (aRR 1.3; 95%CI 0.6–3.1).

When we restricted our study population to the 1,017 hospitalised cases, 202 (20%) were able to be linked to a stay in ICU. The proportion admitted to ICU was 20% (174) for B.1.1.7 cases, 22% (5) for B.1.351 and 21% (23) for non-VOC cases. No difference was found in the risk of ICU admission between B.1.1.7 and non-VOC hospitalised cases in the crude analysis (RR 0.9; 95%CI 0.6–1.4), or after adjusting for sex, age group, country of birth and number of comorbidities (aRR 1.1; 95%CI 0.7–1.6). Similarly, no difference was found in the risk of ICU admission between B.1.351 and non-VOC hospitalised cases in the crude analysis (RR 1.0; 95%CI 0.4–2.4), or after adjusting for sex, age group, country of birth and number of comorbidities (aRR 1.2; 95%CI 0.5–2.6), although the cohort of hospitalised patients infected with B.1.351 was small (n = 23).

## Discussion

In this study, we have analysed individual-level data on all laboratory-confirmed cases of COVID-19 in Norway, hospitalisations and ICU admissions among cases, as well as demographic characteristics and underlying comorbidities. Our findings add to the growing evidence of a higher risk of severe disease among persons infected with B.1.1.7. We estimate a 1.9-fold increased risk of hospitalisation for B.1.1.7 compared to non-VOC. Studies from Denmark (aRR 1.4) [4] and using pooled surveillance data from seven European countries (aOR 1.7) [7] found a similar association. In a sensitivity analysis, where we used a similar definition of hospitalisation to the study from Denmark, our estimate of the aRR remained 1.9 (S1 File, part 3.2). Differences in the magnitude of the association between studies may be due to differences in the setting, definition of a COVID-19 hospitalisation, methods of data collection, analysis and length of follow-up period.

We found that B.1.1.7 infection was associated with higher risk of ICU admission compared to non-VOC, in line with other studies [7, 9]. However, when we restricted our study population to hospitalised cases, we found no difference in the risk of ICU admission between B.1.1.7 and non-VOC. In a separate analysis, we have also found no difference in the time from symptom onset to hospitalisation, length of stay in hospital or ICU, nor odds of mortality up to 30

days post discharge for persons infected with B.1.1.7 compared to non-VOC in Norway [17]. Several studies from the UK have also found no evidence of an association between severe disease, death, and/or need for increased ICU resources among hospitalised patients infected with B.1.1.7, compared to other lineages [9, 18–20]. This suggests that, while B.1.1.7 seems to increase the risk of hospitalisation, other patient characteristics may determine patient trajectories and healthcare required among those hospitalised with COVID-19. We did not have access to necessary variables to explore clinical differences between patients infected with different variants of SARS-CoV-2, and there is a need for further studies on differences in clinical characteristics among patients infected with a VOC.

Our study supports evidence that infection with B.1.351 is associated with increased risk of hospitalisation and ICU admission [7, 12, 13], although more studies are needed to explore this association, particularly among hospitalised cohorts. In our cohort, we did not observe an association between B.1.351 and ICU admission among hospitalised patients, although the sample size was small. The association between B.1.351 and severe disease is concerning, amid reports of reduced effectiveness of the ChAdOx1 nCoV-19 vaccine against this variant [21]. Recent evidence suggests, however, that current mRNA vaccines are effective in preventing infection and severe disease with B.1.351 [22], while a study from South Africa found that the NVX-CoV2373 vaccine is 60% effective in preventing symptomatic B.1.351 infection among HIV-negative adults [23].

Norway is a country with both high testing activity and sequencing capacity. Our results can be generalised considering that 49% of all reported cases had known variant data during the study period, and we did not find any systematic differences between our study cohort and notified cases which influenced our results. Our results were robust when we conducted various sensitivity analyses, such as restricting our study population to only cases sequenced with WGS.

One potential source of bias could be if there are systematic differences between the variants among non-diagnosed cases. For example, if in fact VOC cause more severe disease, a smaller proportion of VOC infections may be asymptomatic or present with mild symptoms. This may lead to a larger proportion of VOC cases being diagnosed compared to non-VOC, which could result in an underestimation of the true RR of hospitalisation. An ecological study from the UK reported no changes in self-reported symptoms or asymptomatic infections, during a period of increasing spread of B.1.1.7 [24], although more studies on this subject are needed. Another potential source of bias is that the testing strategy in Norway was enhanced from mid-February, with more contact tracing and extensive testing during a period when B.1.1.7 and B.1.351 were more prevalent. Therefore, a larger proportion of all cases of these two VOC in the study period may have been diagnosed, compared to non-VOC. This could also result in an underestimation of the risk of hospitalisation for B.1.17 and B.1.351. However, when we restricted our analysis to cases diagnosed from week 7, 2021, our results were robust. Conversely, the increased concern regarding VOC could lead to clinicians hospitalising patients more often, which would cause an overestimation of the relevant RRs. As we did not have access to data on disease severity at admission, we cannot rule out this bias in our analysis.

It should be noted that the reported main cause of hospitalisation is a clinical assessment. We cannot rule out that COVID-19 may have been a contributing factor to admission for some patients reported as having another main cause of hospitalisation. However, there is no reason to believe that this assessment would differ between patients infected with different variants. We also conducted a sensitivity analysis by including cases hospitalised with another main cause as not hospitalised and hospitalised, and our main results were robust. In addition, the method used to determine underlying comorbidities will likely underestimate the true prevalence, as only individuals that have been in contact with health services are identified.

Also, data on medications used and procedure codes are currently not taken into account, which would improve the definitions and detect more individuals with underlying comorbidities.

Both B.1.1.7 and B.1.351 have now spread worldwide. While ongoing vaccination programmes have proved effective in reducing the incidence of infection and severe disease from both VOC and non-VOC strains, the emergence of new variants will remain an area of substantial concern as we continue to battle the spread of SARS-CoV-2. The VOC B.1.617.2 has already outcompeted B.1.1.7 in several countries [2, 25–27]. B.1.617.2 has also been associated with increased risk of hospitalisation compared to B.1.1.7, although vaccines have been reported to be effective in reducing the risk of B.1.617.2 infection and hospitalisation [25, 26]. The findings in this and other studies should stress the importance of prevention and control measures to reduce transmission of VOC in society, particularly ongoing vaccination programmes, and preparedness plans for surge capacity in the hospital sector.

## Supporting information

**S1 File. Supporting data and information as referred to in the manuscript, including details on the identification of cases with underlying comorbidities, the assessment of representativeness of the study population and sensitivity analyses.**
(DOCX)

## Acknowledgments

First and foremost, we wish to thank all those who have helped report data to the national emergency preparedness registry at the Norwegian Institute of Public Health (NIPH) throughout the pandemic. We also highly acknowledge the efforts that regional laboratories have put into establishing a routine variant screening procedure or whole genome sequencing at short notice and registration of all analysis in national registries for surveillance. Thanks also to the staff at the Virology and Bacteriology departments at NIPH involved in national variant identification and whole genome analysis of SARS-CoV-2 viruses. We also highly acknowledge the efforts of staff at hospitals around Norway to ensure the reporting of timely and complete data to the Norwegian Intensive Care and Pandemic Registry, as well as colleagues at the register itself. We would also like to thank Anja Elsrud Schou Lindman, project director for the national preparedness registry, and all those who have enabled data transfer to this registry, especially Gutorm Høgåsen at the NIPH, who has been in charge of the establishment and administration of the registry. We would also like to thank Hanne Gulseth and 'Team risk group' at the NIPH, who developed the data cleaning procedure for underlying comorbidities in the preparedness registry, as well as Trude Marie Lyngstad, Anders Skyrud Danielsen, Nora Dotterud and Evy Dvergsdal at the NIPH for their assistance in cleaning the data from different registries.

## Author Contributions

**Conceptualization:** Eirik Alnes Buanes, Karin Nygård, Robert Whittaker.

**Data curation:** Lamprini Veneti, Elina Seppälä, Margrethe Larsdatter Storm, Beatriz Valcarcel Salamanca, Eirik Alnes Buanes, Nina Aasand, Umaer Naseer, Karoline Bragstad, Olav Hungnes, Håkon Bøås, Reidar Kvåle, Robert Whittaker.

**Formal analysis:** Lamprini Veneti, Elina Seppälä, Beatriz Valcarcel Salamanca, Robert Whittaker.

**Methodology:** Lamprini Veneti, Beatriz Valcarcel Salamanca, Robert Whittaker.

**Project administration:** Robert Whittaker.

**Writing – original draft:** Lamprini Veneti, Elina Seppälä, Margrethe Larsdatter Storm, Robert Whittaker.

**Writing – review & editing:** Lamprini Veneti, Elina Seppälä, Margrethe Larsdatter Storm, Beatriz Valcarcel Salamanca, Eirik Alnes Buanes, Nina Aasand, Umaer Naseer, Karoline Bragstad, Olav Hungnes, Håkon Bøås, Reidar Kvåle, Karan Golestani, Siri Feruglio, Line Vold, Karin Nygård, Robert Whittaker.

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
