## [Decision Letter · Decision Letter 0]

19 Jul 2021

PONE-D-21-21292

Increased risk of hospitalisation and intensive care admission associated with infection with SARS-CoV-2 variants B.1.1.7 and B.1.351 in Norway, December 2020 – May 2021

PLOS ONE

Dear Dr. Whittaker,

Thank you for submitting your manuscript to PLOS ONE. After careful consideration, we feel that it has merit but does not fully meet PLOS ONE’s publication criteria as it currently stands. Therefore, we invite you to submit a revised version of the manuscript that addresses the points raised during the review process.

Please address the issues and revise accordingly.

We look forward to receiving your revised manuscript.

Kind regards,

Academic Editor

PLOS ONE

Journal Requirements:

Reviewers' comments:

Reviewer's Responses to Questions

**Comments to the Author**

1. Is the manuscript technically sound, and do the data support the conclusions?

Reviewer #1: Partly

Reviewer #2: Yes

2. Has the statistical analysis been performed appropriately and rigorously? 

Reviewer #1: No

Reviewer #2: Yes

3. Have the authors made all data underlying the findings in their manuscript fully available?

Reviewer #1: Yes

Reviewer #2: Yes

4. Is the manuscript presented in an intelligible fashion and written in standard English?

Reviewer #1: Yes

Reviewer #2: Yes

5. Review Comments to the Author

Reviewer #1: Several comments,

In the introduction section, the authors should update the current status of SARS-CoV-2 variants, especially the delta variant. Moreover, what is the policy of Norway health system while patients infected with SARS-CoV-2 was identified? Several descriptions are not clear. For example, “The proportion of confirmed cases of COVID-19 with known virus variant…”, this description is unclear and should be specified; “…. the dominating variant since week 7, 2021”, what is the proportion of patients infected with this variant and the data source?

In the result section, severe comments: 1.) the resolution of figure 1 is poor. 2.) from line 196 to line 202, it is unclear about “the proportion of cases in our study….” and “For the higher proportion of cases in our study…”, please makes them clearer. 3.) As the authors described, “the proportion of cases with known variant data increased from 8% in weeks 53/2020–1/2021 to 78% in weeks 8–9 and stayed between 46 to 75% 181 during weeks 10–17”, there is a wide variation in the proportion of patients receiving variant testing. For those cases not receiving testing, especially in the earlier weeks of this study, how many cases were admitted to hospital or ICU?

Finally, two main problems of this study should be clarified. First, how about the disease severity of included patients at presentation. Second, how many days have been passed since the onset of symptoms among the included patients at presentation.

Reviewer #2: This article, entitled “Increased risk of hospitalisation and intensive care admission associated with infection with SARS-CoV-2 variants B.1.1.7 and B.1.351 in Norway, December 2020–May 2021”, described the increased severity of COVID-19 illnesses of VOC strains, B.1.1.7 and B.1.351. The data analysis is stringent and persuasive.

But, some questions were still raised, and listed below:

1. Abstract

i. The authors may clarify the sentence ” Among hospitalised cases, no difference was found in the risk of ICU admission between B.1.1.7 and non-VOC”, which was not parallel with the title of the article.

2. Introduction

3. Methods

i. According to the manuscript, the study population had excluded the persons without national identity number. Please describe the characteristics of COVID-19 confirmed cases who did not have national identity number, like immigrants, foreign workers, or visitors. Did these cases have different risks of hospitalization and intensive care admission?

4. Results

i. The authors had excluded the cases who had received COVID-19 vaccines. Do the authors think that vaccine is a protective factors? Did the authors do sensitivity analysis including the vaccinees?

ii. Please make a correction on the figure legend: Figure 1. Distribution of COVID-19 cases reported and screened with B.1.1.7, B.1.351 and non-VOC by week of sampling, Norway, December 2020 – May 2021. Note: ' All cases reported' (n=60,606) excludes vaccinated reported cases, cases who had a national identity number registered and….

iii. May the authors explain the meanings of the sentences at line 196: The proportion of cases in our study cohort was higher among hospitalised cases than among those not hospitalised (54% vs 46%)…...

iv. Table 2 showed that the aRR was highest among patients infected with B.1.1.7 without comorbidity, compared with non-VOC. Please explain.

5. Discussions

i. Regarding the variants B.1.1.7 and B.1.351, increased rates of hospitalization and ICU admission had been described in Denmark, England, South Africa, and European pooled data. Did the authors consider what the uniqueness of this article?

6. The conclusion of this article “There is a need for further studies on differences in patient trajectories and clinical characteristics among patients infected with VOC in hospitalised cohorts” was not relevant and did not echo with the discussions. The authors may conclude with preparedness for the increasing hospitalization and ICU admission.

6. PLOS authors have the option to publish the peer review history of their article (what does this mean?). If published, this will include your full peer review and any attached files.

Reviewer #1: No

Reviewer #2: **Yes: **Shu-Hsing Cheng

---

## [Author Response · Author response to Decision Letter 0]

31 Jul 2021

Please find all our responses to editors' and reviewers' comments attached as part of our submission.

---

## [Decision Letter · Decision Letter 1]

18 Aug 2021

PONE-D-21-21292R1

Increased risk of hospitalisation and intensive care admission associated with reported cases of SARS-CoV-2 variants B.1.1.7 and B.1.351 in Norway, December 2020 – May 2021

PLOS ONE

Dear Dr. Whittaker,

Thank you for submitting your manuscript to PLOS ONE. After careful consideration, we feel that it has merit but does not fully meet PLOS ONE’s publication criteria as it currently stands. Therefore, we invite you to submit a revised version of the manuscript that addresses the points raised during the review process.

Please revise accordingly.

We look forward to receiving your revised manuscript.

Kind regards,

Academic Editor

PLOS ONE

Journal Requirements:

Reviewers' comments:

Reviewer's Responses to Questions

**Comments to the Author**

1. If the authors have adequately addressed your comments raised in a previous round of review and you feel that this manuscript is now acceptable for publication, you may indicate that here to bypass the “Comments to the Author” section, enter your conflict of interest statement in the “Confidential to Editor” section, and submit your "Accept" recommendation.

Reviewer #1: (No Response)

Reviewer #2: All comments have been addressed

2. Is the manuscript technically sound, and do the data support the conclusions?

Reviewer #1: (No Response)

Reviewer #2: No

3. Has the statistical analysis been performed appropriately and rigorously? 

Reviewer #1: (No Response)

Reviewer #2: Yes

4. Have the authors made all data underlying the findings in their manuscript fully available?

Reviewer #1: (No Response)

Reviewer #2: Yes

5. Is the manuscript presented in an intelligible fashion and written in standard English?

Reviewer #1: (No Response)

Reviewer #2: Yes

6. Review Comments to the Author

Reviewer #1: 1. Regarding the policy of government health system, are all patients infected with SARS-CoV-2 admitted or only those with conditional criteria, such as moderate to severe severity, hospitalized? During the study period, were the standards of hospitalization for patients with SARS-CoV-2 consistent?

2. How about the comparison of the median time from symptom onset to hospitalization between various group? Data about this parameter and others (testing time to hospitalization) between groups should be presented in the article. Over half of patients had no records about the time from symptom onset to hospitalization (line 242 -243). How about the proportion of the missing data among different groups?

3. The decision of hospitalization may depend on objective parameters (disease severity, not available in this study) or subjective parameters (physician personal decision or the policy of healthcare system). If the decision of hospitalization is based on subjective parameter, selection bias may be present. Therefore, it is necessary to clarify the standards of hospitalization.

The English grammar should be edited

Reviewer #2: The authors replied to the comments comprehensively, and had done very detail. I have no more comments.

7. PLOS authors have the option to publish the peer review history of their article (what does this mean?). If published, this will include your full peer review and any attached files.

Reviewer #1: No

Reviewer #2: **Yes: **Shu-Hsing Cheng

---

## [Author Response · Author response to Decision Letter 1]

19 Aug 2021

Please find our responses to reviewers' comments attached as part of our resubmission.

---

## [Decision Letter · Decision Letter 2]

29 Sep 2021

Increased risk of hospitalisation and intensive care admission associated with reported cases of SARS-CoV-2 variants B.1.1.7 and B.1.351 in Norway, December 2020 – May 2021

PONE-D-21-21292R2

Dear Dr. Whittaker,

We’re pleased to inform you that your manuscript has been judged scientifically suitable for publication and will be formally accepted for publication once it meets all outstanding technical requirements.

Kind regards,

Academic Editor

PLOS ONE

Additional Editor Comments (optional):

Reviewers' comments:

Reviewer's Responses to Questions

**Comments to the Author**

1. If the authors have adequately addressed your comments raised in a previous round of review and you feel that this manuscript is now acceptable for publication, you may indicate that here to bypass the “Comments to the Author” section, enter your conflict of interest statement in the “Confidential to Editor” section, and submit your "Accept" recommendation.

Reviewer #2: All comments have been addressed

Reviewer #3: (No Response)

2. Is the manuscript technically sound, and do the data support the conclusions?

Reviewer #2: Yes

Reviewer #3: Yes

3. Has the statistical analysis been performed appropriately and rigorously? 

Reviewer #2: Yes

Reviewer #3: Yes

4. Have the authors made all data underlying the findings in their manuscript fully available?

Reviewer #2: Yes

Reviewer #3: No

5. Is the manuscript presented in an intelligible fashion and written in standard English?

Reviewer #2: Yes

Reviewer #3: Yes

6. Review Comments to the Author

Reviewer #2: The authors had done their best to analyze their data and to clarify the limitation of this study. No further comments to the authors. They had done a great job.

Reviewer #3: This is a very interesting submission by Veneti et al, investigating the risk of hospitalisation and ICU admission following infection with SARS-CoV-2 variants of concern (VOC) B.1.1.7 and B.1.351 compared to non-VOC infections.

This is well written and statistical analysis had been performed to a high standard.

My main concern was that why other VOC were removed from the analysis and why other VOC were not included as comparators. After reviewing Table S2, it is evident that the prevalence of the other VOC was relatively low in this cohort compared to B.1.1.7 and B.1.351. Therefore, I feel that it is justified to use this methodology. Perhaps adding a sentence to this effect within the methods section would add justification to the exclusion of the other VOC.

7. PLOS authors have the option to publish the peer review history of their article (what does this mean?). If published, this will include your full peer review and any attached files.

Reviewer #2: **Yes: **Shu-Hsing Cheng

Reviewer #3: No

---

## [Editor Report · Acceptance letter]

1 Oct 2021

PONE-D-21-21292R2 

Increased risk of hospitalisation and intensive care admission associated with reported cases of SARS-CoV-2 variants B.1.1.7 and B.1.351 in Norway, December 2020 – May 2021  

Dear Dr. Whittaker:

I'm pleased to inform you that your manuscript has been deemed suitable for publication in PLOS ONE. Congratulations! Your manuscript is now with our production department. 

Kind regards, 

on behalf of

Dr. Robert Jeenchen Chen 

Academic Editor

PLOS ONE